# Active Learning for Non-Parametric Regression Using Purely Random Trees

**Jack Goetz**              **Ambuj Tewari**              **Paul Zimmerman**

University of Michigan
Ann Arbor, MI 48109
{jrgoetz, tewaria, paulzim}@umich.edu

## Abstract

Active learning is the task of using labelled data to select additional points to label, with the goal of fitting the most accurate model with a fixed budget of labelled points. In binary classification active learning is known to produce faster rates than passive learning for a broad range of settings. However in regression restrictive structure and tailored methods were previously needed to obtain theoretically superior performance. In this paper we propose an intuitive tree based active learning algorithm for non-parametric regression with provable improvement over random sampling. When implemented with Mondrian Trees our algorithm is tuning parameter free, consistent and minimax optimal for Lipschitz functions.

## 1   Introduction

In this paper we study active learning for regression in the pool setting. In our setup we are given a pool of unlabelled data points and want to build the best model with a fixed number of samples, allowing selection of new points to use labels already obtained. Active learning is motivated by scenarios where the experimenter has control over the data labelling process and where unlabelled points are cheap but labels are expensive.

Our primary motivation comes from computational chemistry, where chemical properties of interest can be computed by solving approximations to the Schrödinger equation. One key property to chemists, the rate of chemical reaction, can be quantified via the activation energy, which controls the rate of reaction as a function of temperature [9]. While calculating the activation energy is expensive, there are a small number of readily available features of the reaction that influence the activation energy. This incentivizes building a metamodel for the activation energy to avoid excessive analysis of undesirable (high activation energy) reactions. Since we are restricted in the number of simulations used to build our metamodel, we want to use the most informative data points. Because chemical reactions are discrete entities, we are restricted to a finite (but often large) pool of reactions, thus requiring pool setting active learning even though we are selecting simulations.

Active learning methods are usually built on top of existing prediction algorithms. Decision trees and forests are a popular class of such predictors due to their simplicity, expressiveness, state-of-the-art performance and tuning parameter free nature. In this paper we focus our attention on purely random trees [4], decision trees built independently of any data, due to their amenability to theoretical analysis. We use a recently proposed version called Mondrian Trees [17], which have been shown to produce trees with many attractive properties such as consistency and minimax optimal rate of convergence for Lipschitz functions [19].

As in some previous work [7], our active learning algorithm will be developed in two stages. First we introduce a simple and intuitive *oracle* querying algorithm for purely random trees which is optimal among a natural class of sampling schemes which includes random sampling (Theorem

4.4). This algorithm is not active but uses statistics of the true joint distribution which are generally unknown. Second we propose an active learning scheme where we first sample passively to estimate the required statistics, and then use those estimates to approximate the oracle algorithm. We show this algorithm is consistent for the oracle algorithm (Theorem 5.1) and behaves well when our labels are normally distributed (Theorem 5.4). Finally we examine the empirical performance of our active learning algorithm to show that benefits, though sometimes modest, can be significant.

## 2 Setting and background

We begin by describing the pool based active learning setting, as well as introducing purely random and Mondrian trees. We have a pool of $m$ data points $\{X_i\}_1^m$, with $X_i \in [0,1]^d$ (rescaling our $X$ as needed) and $X_i \sim p_X$, which are always available to the algorithm. For each $X_i$ we have a corresponding label $Y_i \in \mathbb{R}$ with the relationship $Y_i = f(X_i) + \sigma(X_i)\epsilon_i$ with $\epsilon_i \sim p_\epsilon$ iid, $\epsilon_i \perp X_j \; \forall j, E(\epsilon_i) = 0, \text{Var}(\epsilon_i) = 1, \sigma(X_i) : [0,1]^d \to \mathbf{R}_+$, meaning our noise is the product of a function of $X$ with an independent random variable. We assume the $(X_i, Y_i) = D_i$ have been drawn iid from a joint distribution $p_{X,Y}$. We will assume that $f(x)$ and $\sigma(x)$ are bounded.

Initially none of these $Y_i$ are known to the algorithm. Instead we have the ability to gain access to any of the $Y_i$, and the task is to select $n \ll m$ labels with the goal of building a model with the lowest quadratic risk $E\left[(\hat{f}(X) - f(X))^2\right]$, where the expectation is taken over our test point $X$, the random process which builds our tree and the labelled data we select. Throughout we will assume that our pool is arbitrarily large; in particular we will assume that the marginal density $p_X$ is known, and that there is enough unlabelled data to implement any sampling scheme for selecting $n$ points. We use *active* sampling (or learning) to describe any sampling scheme which samples in multiple batches and uses both $X_i's$ as well as known $Y_i's$ from previous batches when picking points for the next batch. We use passive sampling to denote any sampling scheme which only uses the $X_i$ to pick points, and we use *random* sampling to denote picking the points uniformly at random from our pool (which is the same as sampling from $p_{X,Y}$).

Our active learning method is for purely random trees [4], which are decision trees (or partitions of the space) built using a random process that is independent of the data. We will interchangeably discuss the partition of the space generated by the tree and the leaves of the tree. Let $I_k \in \mathcal{I}$ enumerate the leaves of a tree (partitions of the space), where $k \in \{1...K\}$. We will abuse notation slightly and use the set of partitions $\mathcal{I}$ to denote our tree. These partitions can be used to build regressograms, which make predictions using the average of labelled points within the partition of the test point. With the partitions fixed, the best (in $L_2$) approximation to $f$ which is piece-wise constant on each partition predicts the conditional mean on that partition [14]. We will denote true values and estimates of this approximation using "tilde" and "hat" notation as shown below.

| True best approximation | Estimate of best approximation |
|---|---|
| $\tilde{f}_{\mathcal{I}}(x) = \sum_{k=1}^{K} \mathbf{1}(x \in I_k)\tilde{\beta}_k$ | $\hat{f}_{\mathcal{I}}(x) = \sum_{k=1}^{K} \mathbf{1}(x \in I_k)\hat{\beta}_k$ |
| $\tilde{\beta}_k = E_{p_{X,Y}}[Y\|X \in I_k]$ | $\hat{\beta}_k = \dfrac{1}{\sum \mathbf{1}(X_i \in I_k)} \sum_{X_i \in I_k} Y_i$ |

Our experiments and some results will use particular purely random trees built using the Mondrian Process [17]. The Mondrian Process is a stochastic process for partitioning a hypercube in $\mathbb{R}^d$, a single realization of this process gives a Mondrian Tree. The Mondrian Process iteratively splits existing partitions, and the number of partitions is controlled by a parameter $\lambda$ which, since the Mondrian Process is a generalization of a Poisson Process, is referred to as the *lifetime* parameter. As this parameter increases the number of partitions increases, and the rate at which the number of partitions increase depends on the dimension and size of the hypercube. We will use Mondrian Trees on a fixed domain $[0,1]^d$ with varying lifetime as in [19], which describes how these random partitions are built.

# 3 Related work on Active Learning

The majority of theoretical work in active learning has taken place in binary classification, and there are many approaches which have been studied (see, e.g. [13], [10], [22], [16], [3], [2]). These algorithms are studied under fairly nonrestrictive assumptions (except occasionally requiring a linear classification boundary). It has been shown that for a variety of realistic noise conditions active learning provides a better minimax learning rate than passive learning ([15]).

In contrast the theory for active learning in regression is less well developed. A negative result [23] showed that for a Lipschitz regression function and constant noise variance, the minimax learning rate for active learning was the same as that for passive (up to a constant). Additional assumptions are required to obtain better rates. Such structure includes assumptions of piece-wise constantness of regression function [23], approximation of a non-linear model by a linear one [21], locally varying smoothness [6], well-specified parametric model [8] or heteroskedasticity [11], [7].

While many of these regression methods are able to provide provably better learning rates in terms of $n, d$, they are often tailored for their specific assumptions and may perform poorly if the assumptions do not hold. As a recent summary [18] of numerous flexible but guarantee free methods shows, there is great demand for active learning methods without such stringent conditions. Our active learning algorithm will make very mild assumptions, but the improvement will not be in rates in $n, d$ (since it is known this is not always possible). Rather we will adopt the approach [13] of comparing the sampling generated by our algorithm to an optimal sampling scheme, as well as to random sampling.

## 4 Oracle label querying algorithm

We first describe a simple family of querying algorithms for a fixed purely random tree $\mathcal{I}$ which are not active. In the first two subsections below, we will be implicitly conditioning on the tree $\mathcal{I}$, but will suppress this in the notation.

### 4.1 Generic algorithm

In our generic algorithm family, the tree is built without using any data. So we build the tree first and query based on the tree's structure. We call it an "oracle" algorithm since it requires $p_{X,Y}$.

---

**Algorithm 1:** Generic "oracle" querying algorithm

---

**Input:** Leaves of our tree $\mathcal{I}$, pool of data points $\{X_i\}_{i=1}^m$, label budget $n$ and joint distribution $p_{X,Y}$
**Output:** The set of points to label
**foreach** $I_k \in \mathcal{I}$ **do**
> Calculate $q_k$ the proportion of points to select from leaf $I_k$, using $\mathcal{I}, \{X_i\}_{i=1}^m, n, p_{X,Y}$. ;
> Select $n_k = n \cdot q_k$ points uniformly at random from the pool of unlabelled points in that leaf. ;
**end**

---

The algorithm is described as picking $n_k$ deterministically for simplification of notation in proofs. However it is clear that if the $n_k$ are random then it is easy (in principle) to discuss the probabilistic properties of the algorithm, and the details of the risk under random versions of Algorithm 1 are discussed in the proof for Corollary 4.6. The pool marginal distribution $p_X$ and the proportion in each leaf $q_k$ from the querying algorithm above induce a marginal distribution $p'_X$, as well as a joint distribution $p'_{X,Y} = p_{Y|X}p'_X$. The scheme is very general, and it is worth noting that random sampling is a (randomized) version of Algorithm 1. But this is enough structure to produce a somewhat obvious but very important property of our sampling distribution restricted to each leaf.

**Proposition 4.1.** *Fix a tree structure $\mathcal{I}$, pool marginal density $p_X$ and version of Algorithm 1, giving us an induced marginal density $p'_X$. Let $p'_X(X|I_k) = p'_X(X|X \in I_k)$ denote the induced marginal density conditioned on $X \in I_k$. Then as long as $q_k \neq 0$, $p'_X(X|I_k) = p_X(X|I_k)$ for any version of Algorithm 1.*

One important property this gives us is that $E_{p'_{X,Y}}[\hat{\beta}_k] = \tilde{\beta}_k$ (as long as $I_k$ has at least 1 labelled point to estimate $\hat{\beta}_k$), meaning our sampling scheme produces unbiased estimates of the optimal regressogram for this tree. It also allows for a bias-variance decomposition of the risk of the tree.

This decomposition was already known [12] under the assumption of independence between tree structure and the data. We relax this assumption slightly as the distribution of the data depends on the structure of the tree, but still permits this decomposition.

**Corollary 4.2.** *For a fixed tree structure $\mathcal{I}$, under any sampling distribution generated by Algorithm 1 we have the following bias-variance decomposition of our risk:*

$$E\left[(\hat{f}_{\mathcal{I}}(X) - f(X))^2\right] = E\left[(\tilde{f}_{\mathcal{I}}(X) - f(X))^2\right] + E\left[(\hat{f}_{\mathcal{I}}(X) - \tilde{f}_{\mathcal{I}}(X))^2\right].$$

We will refer to these as the *risk bias term* and *risk variance term.* The risk bias term depends only on the structure of the tree, which does not depend our sampling scheme. We thus focus on the risk variance term. Again using Proposition 4.1 we show this term for a single leaf takes a simple form.

**Lemma 4.3.** *For a fixed tree structure $\mathcal{I}$, under any sampling distribution generated by Algorithm 1 we have that the variance error term on the leaf $I_k$ is:*

$$E\left[(\hat{f}_{\mathcal{I}}(X) - \tilde{f}_{\mathcal{I}}(X))^2 | X \in I_k\right] = \frac{1}{n_k}\left(bias_k^2 + \sigma_{\epsilon,k}^2\right) = \frac{1}{n_k}Var(Y|X \in I_k),$$

$$bias_k^2 := E_{p_{X,Y}}\left[(f(X) - \tilde{\beta}_k)^2 | X \in I_k\right], \quad \sigma_{\epsilon,k}^2 := E_{p_{X,Y}}\left[(\sigma(X)\epsilon)^2 | X \in I_k\right].$$

## 4.2 Optimal algorithm

In the above lemma we have emphasized that the terms $bias_k^2$ and $\sigma_{\epsilon,k}^2$ have expectations taken with respect to the data generating distribution $p_{X,Y}$ and do not depend on the induced distribution $p'_{X,Y}$. Thus the only way our sampling distribution affects the variance term is through $n_k$. Averaging out over the contribution of each leaf we get that our overall variance error term is.

$$E\left[(\hat{f}_{\mathcal{I}}(X) - \tilde{f}_{\mathcal{I}}(X))^2\right] = \sum_k P(X \in I_k)\frac{1}{n_k}\left(bias_k^2 + \sigma_{\epsilon,k}^2\right). \tag{1}$$

Let $p_k = P(X \in I_k)$ under the pool marginal distribution and $\sigma_{Y,k}^2 = bias_k^2 + \sigma_{\epsilon,k}^2$. Now we are given a budget of $n$ data points, and we want to minimize our variance error term subject to this budget. This gives us the following optimization problem which can be easily solved:

$$\begin{aligned} \underset{n_k}{\text{minimize}} \quad & \sum_k \frac{1}{n_k}p_k\sigma_{Y,k}^2 \\ \text{subject to} \quad & \sum_k n_k = n \end{aligned} \quad \rightarrow \quad n_k^* = n\frac{\sqrt{p_k\sigma_{Y,k}^2}}{\sum_{k'}\sqrt{p_{k'}\sigma_{Y,k'}^2}}$$

The proportions are very intuitive; cells with high bias and/or noise, or high (test) marginal density will get more samples. These results are summarized in the following theorem:

**Theorem 4.4.** *Let $Y_i = f(X_i) + \sigma(X_i)\epsilon_i$ and fix the partitions $\mathcal{I}$ of our tree. The risk minimizing oracle querying algorithm out of the family of algorithms described by Algorithm 1 is the one with the following $n_k$ and error*

$$n_k^* = n\frac{\sqrt{p_k\sigma_{Y,k}^2}}{\sum_{k'}\sqrt{p_{k'}\sigma_{Y,k'}^2}}, \qquad E\left[(\hat{f}_{\mathcal{I}}(X) - \tilde{f}_{\mathcal{I}}(X))^2\right] = \frac{1}{n}(\sum_k\sqrt{p_k\sigma_{Y,k}^2})^2.$$

**Definition 4.5.** The distribution induced by the sampling in Theorem 4.4 will be referred to as $p_X^*$.

**Remark.** This has a similar flavour to uncertainty sampling methods from classification in that regions with greater variation will get more samples. However whereas in classification sampling can focus locally near the decision boundary, in regression sampling must remain global.

Random sampling is a randomized version of Algorithm 1, so the risk under random sampling is the bias term plus a weighted average of the variance terms for different $(n_1, ..., n_K)$. The sampling

scheme from Theorem 4.4 has the same bias term, but minimizes the variance term meaning our optimal sampling scheme is better than any randomized version of Algorithm 1 (as long as $m > n$), including random sampling.

**Corollary 4.6.** *For a fixed tree structure $\mathcal{I}$, the risk from any randomized version of Algorithm 1 is greater than the risk from sampling according to $p_X^*$ unless $P(n_1^*, ..., n_K^*) = 1$. In particular sampling according to $p_X^*$ is strictly better than random sampling.*

We can also calculate the excess error if we use the incorrect values of $\sigma_{Y,k}^2$. Let $\tilde{\sigma}_{Y,k}^2 = a_k \sigma_{Y,k}^2$, so $a_k$ is a multiplicative error (we will see that our errors will be multiplicative). Given fixed leaf errors $a_1, ..., a_K$ we can calculate the additional risk generated by using $\tilde{\sigma}_{Y,k}^2$ in our optimal algorithm instead of the true $\sigma_{Y,k}^2$

**Lemma 4.7.** *For a fixed tree structure $\mathcal{I}$, if $n_k = n\dfrac{\sqrt{p_k \tilde{\sigma}_{Y,k}^2}}{\sum\limits_{k'} \sqrt{p_{k'} \tilde{\sigma}_{Y,k'}^2}}$ and the variance error term for each leaf is as in Lemma 4.3, then our risk variance is:*

$$E\left[(\hat{f}_{\mathcal{I}}(X) - \tilde{f}_{\mathcal{I}}(X))^2\right] = \frac{1}{n}(\sum_k \sqrt{p_k \sigma_{Y,k}^2})^2 + \frac{1}{n}\sum_{k<l}(\frac{\sqrt{a_k}}{\sqrt{a_l}} + \frac{\sqrt{a_l}}{\sqrt{a_k}} - 2)\sqrt{p_k p_l \sigma_{Y,k}^2 \sigma_{Y,l}^2}$$

$$:= \text{OPT} + \text{EXCESS}.$$

This also lets us get a sense for the suboptimality of random sampling. If we let $a_k = \frac{p_k}{\sigma_{Y,k}^2}$ then we get $n_k = np_k$ which is the expected number of samples per leaf under random sampling, and so for large $n$ the calculated EXCESS term will be close to the excess risk under random sampling. This gives us the following excess error, which can be small (or even zero) as expected since random sampling can be near-optimal. But if there is varying $Y$ variance across the space this can be large:

**Corollary 4.8.** *For a fixed tree structure $\mathcal{I}$ let $a_k = \frac{p_k}{\sigma_{Y,k}^2}$. Then our excess error is:*

$$\text{EXCESS} = \frac{1}{n}\sum_{k<l}(\sqrt{p_k \sigma_{Y,l}^2} - \sqrt{p_l \sigma_{Y,k}^2})^2 \leq \frac{K}{n}\max_k \sigma_{Y,k}^2.$$

### 4.3 Additional results using Mondrian Trees

The above results hold for any purely random tree. We will now not assume that $\mathcal{I}$ is fixed, but is randomly built using the Mondrian Process and will take expectation over the tree building process as well. Mondrian Trees trained using random sampling are minimax optimal for Lipschitz regression functions when the sequence of lifetime parameters satisfy $\lambda_n \asymp n^{1/(d+2)}$ and $\text{Var}(Y) < \infty$ [19]. Additionally Mondrian Trees with random sampling are weakly universally consistent under the same lifetime sequence and variance assumption. Since the optimal oracle algorithm has smaller risk we immediately get minimax optimal rates in terms of $n, d$ under the same assumptions lifetime sequence by Proposition 4 in [19] and Theorem 4.4, and weak consistency under Theorem 1 in [20].

**Corollary 4.9.** *Let our purely random trees be Mondrian Trees with lifetime parameters $\lambda_n \asymp n^{1/(d+2)}$, and let $Y = f(X) + \sigma(X)\epsilon$, $\text{Var}(Y) < \infty$. If our training data is sampled according to $p_X^*$ then the resulting regressogram has (as $n, m \to \infty$) minimax optimal rates, in terms of $n, d$, over Lipschitz functions with $E\left[(\hat{f}(X) - f(X))^2\right] = \mathcal{O}(n^{\frac{-2}{2+d}})$ and is weakly consistent.*

## 5 Active learning algorithm

The oracle querying algorithm has many appealing qualities. However it requires knowledge of the $\sigma_{Y,k}^2$ which are never known in practice. In this section we propose a two stage active "oracle estimating" algorithm to remedy this deficiency. In our first stage we sample $n_{(1)}$ points according to Algorithm 1 and use those samples to calculate estimates $\hat{\sigma}_{Y,k}^2$ of $\sigma_{Y,k}^2$, which in turn produce estimates $\hat{n}_k$ of $n_k^*$. In the second stage we sample $n_{(2)} = n - n_{(1)}$ points such that the total number of samples in each leaf are $\hat{n}_k$. We analyze the consequences of using these estimates, and show that in the case when $Y$ are normal, our trees are Mondrian Trees, and our Stage 1 samples equally in each leaf, our active algorithm is eventually near optimal with high probability. We also show that

in general this algorithm's estimates $\hat{n}_k$ are consistent for $n_k^*$. Below we give the active algorithm. By using this algorithm we have introduced two complications: One is the estimates will have errors from using estimates $\hat{\sigma}_{Y,k}^2$. The other comes from reusing the data from Stage 1 in our estimates of $\hat{\beta}_k$. Since active learning is used exactly when data is difficult to label, to make an algorithm which is practically appealing it is important to make the most out of any labelled data. However this introduces dependency between $\hat{\beta}_k$ and $\hat{n}_k$. These issues will each be addressed separately.

---

**Algorithm 2:** Active "oracle estimating" algorithm

---

**Input:** Leaves of our tree $\mathcal{I}$, pool of data points $\{X_i\}_{i=1}^m$, and label budgets
$\quad\quad n_{(1)}, n_{(2)}, n = n_{(1)} + n_{(2)}$.
**Output:** The set of labelled points.
*Stage 1* ;
Query $n_{(1)}$ data points using a version of Algorithm 1. ;
Use those samples $(X_i, Y_i)$ to estimate $\hat{\sigma}_{Y,k}^2 = \frac{1}{n_{(1),k}-1} \sum\limits_{X_i \in I_k} (\hat{\beta}_{(1),k} - Y_i)^2$ for each leaf. ;

*Stage 2* ;
**foreach** $I_k \in \mathcal{I}$ **do**

$\quad$ Calculate $\hat{n}_k = n \dfrac{\sqrt{p_k \hat{\sigma}_{Y,k}^2}}{\sum\limits_{k'} \sqrt{p_{k'} \hat{\sigma}_{Y,k'}^2}}$ the number of points in the leaf to sample. ;

$\quad$ Select uniformly at random $n_{(2),k}$ points to query from the leaf so the number of points is $\hat{n}_k$. ;
**end**

---

## 5.1 Using estimates of $n_k^*$

First we analyze (as $n$ increases) the effect of using the estimates $\hat{\sigma}_{Y,k}^2$. Let us fix a sequence of trees $\mathcal{I}_{(n)}, |\mathcal{I}_{(n)}| = K_n$. Typically our trees will contain more partitions as we get more data. For a given tree we can estimate the required leaf variances unbiasedly using the standard unbiased sample variance on each leaf $\hat{\sigma}_{Y,k}^2$. Therefore as long as our leaf kurtosis $\kappa_{Y,k} = \frac{\sigma_{Y,k}^4}{(\sigma_{Y,k}^2)^2}$ (and thus the variance of our sample variance) are all finite, and asymptotically our sample variances on each leaf are consistent for the true variances on each leaf, our estimates $\hat{n}_k \to n_k^*$. We require strong consistency of our variance estimates as a function of both our partitioning method and Stage 1 sampling method, which gives us $\hat{n}_k \to n_k^*$ almost surely. If our trees are grown according to a random process then this strong consistency may be depend on attributes of the tree which my only be true in probability, and in this case we get $\hat{n}_k \to n_k^*$ in probability. Both cases are covered in the below theorem, where generally the $b_n$ denote statistics of the tree and $B$ is either $0$ or $\infty$.

**Theorem 5.1.** *Assume $\kappa_{Y,k} < \infty \ \forall \ k, n$, and our sequence of trees $\mathcal{I}_{(n)}$ and Stage 1 sampling algorithm is strongly consistent for estimating the conditional variance $E[(Y - f(X))^2 | X = x]$ as some statistic $b_n \to B$. Then if $b_n \to B$ almost surely our estimates $\hat{n}_k \to n_k^*$ almost surely and if $b_n \to B$ in probability our estimates $\hat{n}_k \to n_k^*$ in probability.*

**Remark.** Note that the condition $\kappa_{Y,k} < \infty \ \forall \ k, n$ is met if $f, \sigma(X)$ are bounded and $\kappa_\epsilon < \infty$.

Now let our sequence of trees be randomly built Mondrian Trees. If we again use $\lambda_n \asymp n^{1/(d+2)}$, as long as $n_{(1)}$ increases linearly with $n$, these conditions are met when our first round of sampling entails sampling equally in each leaf.

**Corollary 5.2.** *Let our purely random trees be Mondrian Trees with lifetime parameter sequence $\lambda_n \asymp n^{1/(d+2)}$ and let $n_{(1)} = cn$, $c \in (0,1)$ a constant. Additionally let Stage 1 query by $n_{(1),k} = \frac{n_{(1)}}{K_n} \ \forall \ k$. If $\kappa_{Y,k} < \infty \ \forall \ k, n$ and $p_X$ is bounded away from 0 and $\infty$ on it's support, so when $p_X > 0$ there exists $c, C$ s.t. $c \le p_X \le C$, then our estimates $\hat{n}_k \to n_k^*$ in probability.*

Even with consistency our finite sample estimates will give us some error in $\hat{n}_k$. The variance of our sample variance is $\text{Var}(\hat{\sigma}_{Y,k}^2) = \frac{1}{n_k}(\sigma_{Y,k}^4 - (\sigma_{Y,k}^2)^2) + \mathcal{O}(\frac{1}{n_k^2}) \approx \frac{1}{n_k}(\kappa_{Y,k} - 1)(\sigma_{Y,k}^2)^2$, so our errors will scale multiplicatively with $\sigma_{Y,k}^2$ when our kurtosis $\kappa_{Y,k}$ are bounded. This allows us to use Lemma 4.7 to bound our excess error given bounds on the (multiplicative) error $a_k = \hat{\sigma}_{Y,k}^2 / \sigma_{Y,k}^2$.

## 5.2 Reusing data

Since we are using the data in Stage 1 both to estimate $\hat{n}_k$ as well as in our estimator $\hat{\beta}_k$, we have introduced dependence between the estimated optimal leaf sample size $\hat{n}_k$ and leaf mean estimate contribution from Stage 1. To understand the effects of this dependence we will break up our estimates of the leaf mean as $\hat{\beta}_k = \frac{n_{(1),k}\hat{\beta}_{(1),k}+n_{(2),k}\hat{\beta}_{(2),k}}{n_{(1),k}+n_{(2),k}}$, where $n_{(i),k}, \hat{\beta}_{(i),k}$ are the number and mean estimate during sampling round $i \in \{1,2\}$. By writing our final mean estimate in terms of our stage-wise mean estimates we can find an expression for this dependence.

**Lemma 5.3.** *For a fixed tree structure $\mathcal{I}$, under Algorithm 2 the risk variance term becomes:*

$$E[(\hat{\beta}_k - \tilde{\beta}_k)^2] = E_{n_{(2),k}}\left[\frac{n_{(1),k}^2}{(n_{(1),k}+n_{(2),k})^2}E_{D_{1:n_{(1)}}}\left[(\hat{\beta}_{(1),k}-\tilde{\beta}_k)^2|n_{(2),k}\right] + \frac{n_{(2),k}\sigma_{Y,k}^2}{(n_{(1),k}+n_{(2),k})^2}\right].$$

The term $E_{D_{1:n_{(1)}}}\left[(\hat{\beta}_{(1),k} - \tilde{\beta}_k)^2|n_{(2),k}\right]$ quantifies the dependency introduced by reusing the samples from $n_{(1)}$. The dependency is between the variance of part of our mean estimators $(\hat{\beta}_{(1),1}, ..., \hat{\beta}_{(1),k})$ and $(n_{(2),1}, ..., n_{(2),K}) = g(\hat{\sigma}_{Y,1}^2, ..., \hat{\sigma}_{Y,K}^2)$. When $\hat{\beta}_{(1),k} \perp n_{(2),k}$ we get back our risk variance term from Lemma 4.3. However when there is dependence we no longer have that the $n_k^*$ from Theorem 4.4 are optimal over algorithms with an active stage as in Algorithm 2, since the optimal $n_k$ will depend on the sampling during Stage 1. This dependency can be complex and is generally unknown, though as long as the effect is not too large the $n_k^*$ will still provide a very good solution, and the $n_k^*$ are still better than random sampling. It is worth noting that our active algorithm can take advantage of this dependency in some cases to outperform Algorithm 1, and we informally discuss this in the appendix.

## 5.3 The Normal case

The complications above depend on the distribution of $a_k = \frac{\hat{\sigma}_{Y,k}^2}{\sigma_{Y,k}^2}$ and the function $g$, which in general are extremely complicated and hard to analyze for arbitrary $f, p_\epsilon, p_X$. However in the case where $Y$ are normally distributed these become tractable.

**Theorem 5.4.** *Let $Y \sim N(\mu(X), \sigma^2(X))$ and $X$ queried according to Algorithm 2 for a fixed tree $\mathcal{I}$. Then the risk variance term for a leaf is as in Lemma 4.3 and we have that with probability at least $1 - \sum_{k=1}^{K} e^{-\frac{(n_{(1),k}-1)\alpha^2}{8}}$ the excess error is bounded by:*

$$\text{EXCESS} \leq \frac{1}{n}\sum_{k<l}\left[\left(\frac{1+\alpha}{1-\alpha}\right)^{1/4} - \left(\frac{1-\alpha}{1+\alpha}\right)^{1/4}\right]^2 \sqrt{p_k p_l \sigma_{Y,k}^2 \sigma_{Y,l}^2}.$$

*Additionally if our trees are a sequence of Mondrian Trees with lifetime parameter sequence $\lambda_n \asymp n^{1/(d+2)}$ and our Stage 1 sampling procedure is to sample equally in each leaf with $n_{(1)} = cn$, $c \in (0,1)$ a constant, then the above bound occurs with probability at least $1 - \delta_1 - \delta_2$ where*

$$\delta_1 = \frac{(1+n^{1/(d+2)})^d}{n^{(d+1)/(d+2)}} \quad \delta_2 = n^{(d+1)/(d+2)} \exp\left(\frac{-\alpha^2}{8}((cn)^{1/(d+2)}) - 1\right).$$

First, note that a larger $n$ allows us to choose a smaller $\alpha$ and the bound on excess error goes to 0 as $\alpha \to 0$. Second, even for the normal case, $d$ large requires a very large $n$ before we get any control on the error probability $\delta_2$. This is consistent with the empirical observation that Mondrian Trees struggle with large $d$.

Finally we also note that there are many reasons why in practice it is impossible to use the exact $n_k^*$. These include the fact that usually $n_k^*$ will be fractional, a leaf may not have $n_k^*$ points in it, or when using the active algorithm $n_{(1),k} > \hat{n}_k$. These issues will be less significant as $n \to \infty$ and we discuss how each is dealt with in the appendix.

# 6   Simulations and experiments

We now examine the benefits of active learning on both simulated and real world data. We simulate 2 data sets, one with differing noise variance (our $\sigma_{\epsilon,k}^2$ term), the other with differing function complexity (our $bias_k^2$ term), in different regions of $[0,1]^d$. We also examine performance on the Wine quality data set from UCI and a data set of activation energies of Claisen rearrangement reactions (Cl). We compare the performance of selecting points to label using random sampling, our active algorithm, and a naive uncertainty sampling version of our active algorithm, where each leaf $n_k$ is proportional its variance. In all experiments $n_{(1)} = \frac{n}{2}$ and Mondrian Trees are grown using $\lambda_n = n^{\frac{2}{2+d}} - 1$, which is theoretically motivated, but corrected so when $n = 1, \lambda_n = 0$. We use both Mondrian and Breiman Trees [5] as our final regressor. Details of the data sets are in the appendix, which also contains forest versions of these experiments. Additionally all code and experiments (as well as other experiments) are available at `https://github.com/jackrgoetz/Mondrian_Tree_AL`.

When using Mondrian Trees as the final regressor, the active learning method always provides some improvement, and in the simulations this improvement persists when using Breiman Trees. Additionally the uncertainty sampling method sometimes produces worse sampling than random sampling, which is common for direct translations of classification active learning methods. In the real data our benefits are less pronounced, with active learning even being slightly harmful when used with Breiman Trees (although with forests the active learning is beneficial). We believe this performance drop may be due to the inability of the Mondrian Tree to adapt to differing variable importance. It is also possible that our assumptions that $Y$ has changing variance does not hold, and even here the active algorithm is not harmful, where as the naive uncertainty sampling algorithm can be detrimental.

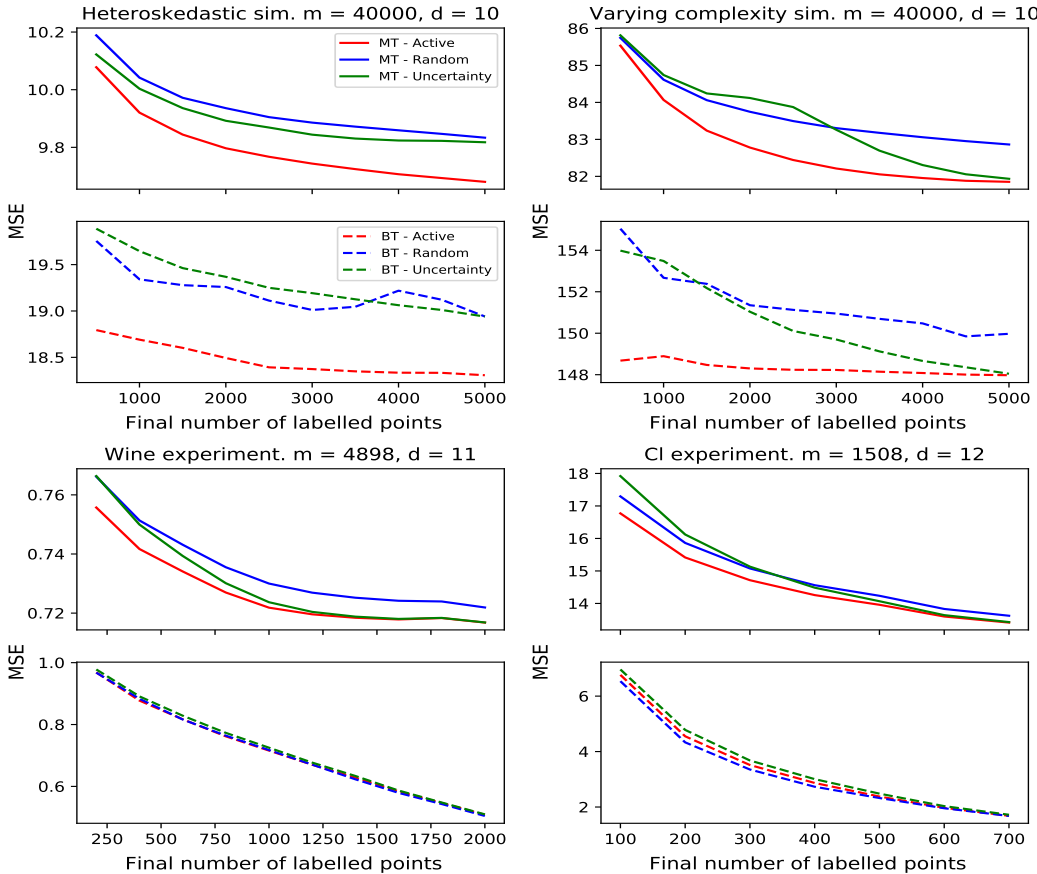

Figure 1: Active learning experiments

# 7 Conclusion and further directions

In this paper we provide a theoretically justified active learning method for non-parametric regression which can take advantage of beneficial structure when present without being detrimental when such structure is absent. When used with Mondrian Trees the method requires no tuning parameters (which are difficult to tune while actively sampling [1]), is asymptotically minimax optimal for Lipschitz regression functions, and is consistent. Although the improvement for active learning in regression is often restricted to constant factor improvements, these constant improvements are important in real world applications.

Despite technical theoretical arguments needed for the theory, the method itself is simple, leading to many interesting avenues for further exploration. One direction would be extending theory to ensembles of trees, or developing tools to deal with high dimensions. Another possibility is to exploit the online nature of Mondrian Trees to develop a parallel theory for streaming based active learning. Finally it may be possible to extend the ideas here to non tree based active learning for regression.

**Acknowledgements**

JG acknowledges the support of NSF via grant DMS-1646108. AT acknowledges the support of a Sloan Research Fellowship.

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
