[Supplementary Material]

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

# 8 Appendix

## 8.1 Proof of Proposition 4.1

This results is nothing more than the fact that a random subsample of size $n < m$ from an initial sample of sizer $m$ has the same distribution as a sample of size $n$ from that original distribution. The only issue here is if $q_k = 0$, in which case $p'_X(x) = 0 \; \forall \; x \in I_k$, where as $p_X(x)$ may be non-zero on a set of positive measure.

## 8.2 Proof of Corollary 4.2

We start by confirming that $E_{p'_{X,Y}}[\hat{\beta}_k] = \tilde{\beta}_k$. Let us fix $\mathcal{I}, k$ with $n$ labelled points and let $n_k = \sum_{i=1}^{n} \mathbf{1}(X_i \in I_k)$. By assumption $n_k > 0$ otherwise $\hat{\beta}_k = \frac{1}{\sum \mathbf{1}(X_i \in I_k)} \sum_{X_i \in I_k} Y_i$ is undefined. Since Algorithm 1 is not active we have that $Y|X \in I_k \perp n_k$.

$$
\begin{aligned}
E_{p'_{X,Y}}[\hat{\beta}_k] &= E_{n_k} E_{p'_{X,Y}}\Big[ \frac{1}{\sum \mathbf{1}(X_i \in I_k)} \sum_{i=1}^{n} Y_i \mathbf{1}(X_i \in I_k)|n_k \Big] \\
&= E_{n_k} \frac{1}{n_k} \sum_{i=1}^{n} E_{p'_{X,Y}}\big[ Y_i \mathbf{1}(X_i \in I_k)|n_k \big] \\
&= E_{n_k} \frac{1}{n_k} \sum_{i=1}^{n} P(X_i \in I_k|n_k) E_{p'_{X,Y}}\big[ Y_i|n_k, X_i \in I_k \big] \\
&= E_{n_k} \frac{1}{n_k} E_{p_{X,Y}}\big[ Y|X \in I_k \big] \sum_{i=1}^{n} P(X_i \in I_k|n_k) \\
&= E_{n_k} E_{p_{X,Y}}\big[ Y|X \in I_k \big] = E_{p_{X,Y}}\big[ Y|X \in I_k \big].
\end{aligned}
$$

Now we use this to derive the decomposition in the standard way.

$$
\begin{aligned}
E\Big[ (\hat{f}_{\mathcal{I}}(X) - f(x))^2 \Big] &= E\Big[ (\hat{f}_{\mathcal{I}}(X) - \tilde{f}_{\mathcal{I}}(X))^2 \Big] + E\Big[ (\tilde{f}_{\mathcal{I}}(X) - f(X))^2 \Big] \\
&\quad + 2 E\Big[ (\hat{f}_{\mathcal{I}}(X) - \tilde{f}_{\mathcal{I}}(X))(\tilde{f}_{\mathcal{I}}(X) - f(X)) \Big].
\end{aligned}
$$

$$
\begin{aligned}
E\Big[ (\hat{f}_{\mathcal{I}}(X) - \tilde{f}_{\mathcal{I}}(X))(\tilde{f}_{\mathcal{I}}(X) - f(X)) \Big] = \\
E[\hat{f}_{\mathcal{I}}(X)] \tilde{f}_{\mathcal{I}}(X) - E[\hat{f}_{\mathcal{I}}(X)] f(X) - \tilde{f}_{\mathcal{I}}(X)^2 + \tilde{f}_{\mathcal{I}}(X) f(X) = 0.
\end{aligned}
$$

## 8.3 Proof of Lemma 4.3

We fix $n_k$. Given $X \in I_k$ we know that $\hat{f}_{\mathcal{I}}(X) = \hat{\beta}_k$ and $\tilde{f}_{\mathcal{I}}(X) = \tilde{\beta}_k$. Let us reorder the data $D_{1:n}$ so that the first $n_k$ are in the leaf $k$ for ease of notation. Then use Proposition 4.1, where the cross term disappears since $\epsilon_i \perp X_i$ under $p_{X,Y}$ by assumption.

$$E_{p'_{X,Y}}\left[(\hat{f}_{\mathcal{I}}(X) - \tilde{f}_{\mathcal{I}}(X))^2 | X \in I_k\right] =$$

$$\frac{1}{n_k^2}\left(\sum_{i=1}^{n_k} E_{p'_{X,Y}}\left[(f(X_i) - \tilde{\beta}_k)^2 | X_i \in I_k\right] + \sum_{i=1}^{n_k} E_{p'_{X,Y}}\left[(\sigma(X_i)\epsilon_i)^2 | X_i \in I_k\right]\right.$$

$$+ 2\sum_{i=1}^{n_k} E_{p'_{X,Y}}\left[(f(X_i) - \tilde{\beta}_k)\sigma(X_i)\epsilon_i | X_i \in I_k\right]\right)$$

$$= \frac{1}{n_k^2}\left(\sum_{i=1}^{n_k} E_{p_{X,Y}}\left[(f(X_i) - \tilde{\beta}_k)^2 | X_i \in I_k\right] + \sum_{i=1}^{n_k} E_{p_{X,Y}}\left[(\sigma(X_i)\epsilon_i)^2 | X_i \in I_k\right]\right)$$

$$= \frac{1}{n_k}\left(bias_k^2 + \sigma_{\epsilon,k}^2\right).$$

### 8.4 Proof of Corollary 4.6

The proof involves looking at the expected risk under a random version of Algorithm 1. Formally allow Algorithm 1 to generate the $q_i$ in a randomized fashion (with the randomness independent from all other sources of randomness), potentially using the other inputs to Algorithm 1 $(\mathcal{I}, \{X_i\}_{i=1}^m, n, p_{X,Y})$ as parameters. Thus $(q_1, ..., q_K)$ are drawn from a distribution, which in turn for all $(n_1, ..., n_K) \in \mathbb{N}^K$ s.t. $\sum n_k = n$ generates $P(n_1, ..., n_K)$ the probability of the algorithm sampling $(n_1, ..., n_K)$ points from each of the tree leaves. Let $Risk(n_1, ..., n_K)$ denote the risk when our by leaf samples sizes are $n_1, ..., n_k$, with $RiskBias$ and $RiskVar(n_1, ..., n_K)$ being the bias and variance terms of the decomposition. The $RiskBias$ does not depend on $n_1, ..., n_K$ since the risk bias term does not depend on how we sample. Then the risk of the randomized version of Algorithm 1 is

$$Risk = \sum_{(n_1,...,n_K)} P(n_1, ..., n_K) Risk(n_1, ..., n_K)$$

$$= RiskBias + \sum_{(n_1,...,n_K)} P(n_1, ..., n_K) RiskVar(n_1, ..., n_K).$$

If $n_1^*, ..., n_K^*$ is our optimal solution then by Theorem 4.4 $RiskVar(n_1^*, ..., n_K^*) \leq RiskVar(n_1, ..., n_K) \; \forall \; (n_1, ..., n_K) \in \mathbb{N}^K$ s.t. $\sum n_k = n$. For random sampling, unless $P(n_1^*, ..., n_K^*) = 1$ the Risk will clearly be greater than (or equal to) that of the optimal since the probability weighted average is greater than (or equal to) the min term of the sum.

### 8.5 Proof of Lemma 4.7

This is all algebra. By Equation 1

$$E\left[(\hat{f}_{\mathcal{I}}(X) - \tilde{f}_{\mathcal{I}}(X))^2\right] = \frac{1}{n}\sum_{k=1}^K \sqrt{a_k}\sqrt{p_k\sigma_{Y,k}^2} \times \sum_{l=1}^K \frac{1}{\sqrt{a_l}}\sqrt{p_l\sigma_{Y,l}^2}$$

$$= \frac{1}{n}\left(\sum_k p_k\sigma_{Y,k}^2 + \sum_{k\neq l} \frac{\sqrt{a_k}}{\sqrt{a_l}}\sqrt{p_kp_l\sigma_{Y,k}^2\sigma_{Y,l}^2}\right)$$

$$= \frac{1}{n}\left(\sum_k p_k\sigma_{Y,k}^2 + \sum_{k<l}(\frac{\sqrt{a_k}}{\sqrt{a_l}} + \frac{\sqrt{a_k}}{\sqrt{a_l}})\sqrt{p_kp_l\sigma_{Y,k}^2\sigma_{Y,l}^2}\right)$$

$$= \frac{1}{n}(\sum_k \sqrt{p_k\sigma_{Y,k}^2})^2 + \frac{1}{n}\sum_{k<l}(\frac{\sqrt{a_k}}{\sqrt{a_l}} + \frac{\sqrt{a_l}}{\sqrt{a_k}} - 2)\sqrt{p_kp_l\sigma_{Y,k}^2\sigma_{Y,l}^2}$$

$$= \text{OPT} + \text{ERROR}.$$

## 8.6 Proof of Corollary 4.8

Again, this is just algebra.

$$\frac{1}{n}\sum_{k<l}\left(\frac{\sqrt{p_k\sigma_{Y,l}^2}}{\sqrt{p_l\sigma_{Y,k}^2}}+\frac{\sqrt{p_l\sigma_{Y,k}^2}}{\sqrt{p_k\sigma_{Y,l}^2}}-2\right)\sqrt{p_kp_l\sigma_{Y,k}^2\sigma_{Y,l}^2}=\frac{1}{n}\sum_{k<l}(\sqrt{p_k\sigma_{Y,l}^2}-\sqrt{p_l\sigma_{Y,k}^2})^2$$

$$\leq\frac{1}{n}\sum_{k<l}(2p_k\sigma_{Y,l}^2+2p_l\sigma_{Y,k}^2)\leq\frac{1}{n}\max_k\sigma_{Y,k}^2\sum_{k\neq l}(p_k+p_l)\leq\frac{K}{n}\max_k\sigma_{Y,k}^2.$$

## 8.7 Proof of Theorem 5.1

By the assumption that our sequence of trees $\mathcal{I}_{(n)}$ and Stage 1 sampling algorithm is strongly consistent for estimating the conditional variance $E[(Y-f(X))^2|X=x]$ as some statistic $b_n\to B$ we have that $\hat{\sigma}_{1,k}^2\to\sigma_k^2$ a.s. as $b_n\to B$. To see this let $\hat{\sigma}_{1,k}^2(x)=\hat{\sigma}_{1,k}^2$ for $x\in I_k$, $\sigma_k^2(x)=\sigma_k^2$ for $x\in I_k$ and let $\sigma^2(x)=E[(Y-f(X))^2|X=x]$. Then $|\hat{\sigma}_{1,k}^2(x)-\sigma_k^2(x)|\leq|\hat{\sigma}_{1,k}^2(x)-\sigma^2(x)|+|\sigma_k^2(x)-\sigma^2(x)|\to 0$, where the first term disappears due to the strong consistency, and the second term disappears due to the size of the partitions shrinking.

If $\hat{\sigma}_{1,k}^2\to\sigma_k^2$ a.s. then $\sum_{k=1}^{K_n}\sqrt{p_k\hat{\sigma}_{1,k}^2}\to\sum_{k=1}^{K_n}\sqrt{p_k\sigma_k^2}$ a.s. as $b_n\to B$. So if $b_n\to B$ a.s. then $\hat{n}_k\to n_k^*$ almost surely.

Now assume $b_n\to B$ in probability as $n\to\infty$ and want to show that these implies $\sum_{k=1}^{K_n}\sqrt{p_k\hat{\sigma}_{1,k}^2}\to$ $\sum_{k=1}^{K_n}\sqrt{p_k\sigma_k^2}$ in probability $n\to\infty$. We will use Lemma 6.3.1.b from [21] which states:

**Lemma** (6.3.1.b in [21])**.** $X_n\to X$ *in probability iff for each subsequence* $\{X_{n_k}\},n_k\to\infty$ *there exists a further subsubsequence* $\{X_{n_{k_t}}\},n_{k_t}\to\infty$ *which converges a.s. to $X$.*

(The $n_k$ here are unrelated to the $n_k$ in our trees).

Let $Y_n=|\sum_{k=1}^{K_n}\sqrt{p_k\hat{\sigma}_{1,k}^2}-\sum_{k=1}^{K_n}\sqrt{p_k\sigma_k^2}|$, so $Y_n\to 0$ a.s. if $b_n\to B$. Thus we have a subset of the overall probability space $\Omega$ which is

$$\Omega\supset\Omega^*=\{\omega\in\Omega:\lim b_n(\omega)\neq B\text{ or }Y_n(\omega)\to 0\}$$

where $P(\Omega^*)=1$. Now take a subsequence $n_k\to\infty$ of $n$. By $b_n\to B$ in probability $\exists n_{k_t}\to\infty$ such that $b_{n_{k_t}}\to B$ a.s. as $n_{k_t}\to\infty$. This gives us a second subset of $\Omega$

$$\Omega\supset\Omega'=\{\omega\in\Omega:b_{n_{k_t}}(\omega)\to B\}$$

where again $P(\Omega')=1$. On the intersection of these we get

$$\Omega^*\cap\Omega'\subset\{\omega\in\Omega:Y_{n_{k_t}(\omega)}(\omega)\to 0\}$$

where $P(\Omega^*\cap\Omega')=1$. $n_k$ was an arbitrary subsequence of $n$ and so by using Lemma 6.3.1.b in the reverse direction we get that $Y_n\to 0$ in probability.

## 8.8 Proof of Corollary 5.2

Here our $b_n=\frac{K_n}{n^{\frac{d+1}{d+2}}}$ and $B=0$. Since $E[K_n]=(1+n^{\frac{1}{d+2}})^d$ by Markov $\frac{K_n}{n^{\frac{d+1}{d+2}}}\to 0$ in probability.

Now we need to show that if we assume $\frac{K_n}{n^{\frac{d+1}{d+2}}} \to 0$ we get strong consistency of our conditional variance function estimation. By Theorem 23.3 in [14] we get that our tree is strongly consistent for estimating the mean function, since $\frac{K_n \log(n)}{n} \to 0$ so eventually every partition will have more than $\log(n)$ samples in the leaf, and the augmented estimator in Theorem 23.3 is the same as the usual estimator. (The augmented estimator in Theorem 23.3 is the usual decision tree estimator if there are more than $\log(n)$ data points in the partition and 0 otherwise). Finally we need the $p_X$ bounded since Theorem 23.3 assumes that our test $X$ density is the same as our training one, but since $p_X$ is bounded the Radon Nikodym derivative is bounded and so we get strong consistency even with the different test density.

So our tree and Stage 1 sampling scheme are strongly consistent for estimating the mean function $f(x) = E[Y|X = x]$. Now assume we had access to a new set of random variables $Z_i = (Y_i = f(X_i))^2$. Because of the bounded kurtosis our tree would also be strongly consistent for estimating the mean function of the $Z_i$ which we will call $f_Z(x) = E[(Y - f(X))^2|X = x]$. So if we had access to the $Z_i$ we could use them to estimate our $Y$ conditional variance function using $\hat{f}_Z(x) = \frac{\sum Z_i \mathbf{1}_{X_i \in I(x)}}{\sum \mathbf{1}_{X_i \in I(x)}}$.

We don't have these $Z_i$ but we do have $\tilde{Z}_i = (Y_i - \hat{f}(X_i))^2$, and it's easy to show that $\frac{\sum \tilde{Z}_i \mathbf{1}_{X_i \in I(x)}}{\sum \mathbf{1}_{X_i \in I(x)} - 1} \to \frac{\sum Z_i \mathbf{1}_{X_i \in I(x)}}{\sum \mathbf{1}_{X_i \in I(x)}}$ by adding and subtracting $f(x)$ inside the square. This gives us a strongly consistent estimator of our conditional variance as required.

## 8.9 Proof of Lemma 5.3

Since the Stage 1 sampling uses Algorithm 1 our $n_{(1),k}$ are fixed (though this could be extended to randomized version of Algorithm 1). The proof is mostly algebra, using the fact that $\hat{\beta}_{(1),k}$ is conditionally independent of $\hat{\beta}_{(2),k}$ given $n_{(2),k}$.

$$
\begin{aligned}
E[(\hat{\beta}_k - \tilde{\beta}_k)^2] &= E\Big[\big(\frac{n_{(1)}(\hat{\beta}_{(1),k} - \tilde{\beta}_k)}{n_{(1),k} + n_{(2),k}} + \frac{n_{(2)}(\hat{\beta}_{(2),k} - \tilde{\beta}_k)}{n_{(1),k} + n_{(2),k}}\big)^2\Big] \\
&= E_{n_{(2),k}}\Big[\frac{n_{(1),k}^2}{(n_{(1),k} + n_{(2),k})^2}E_{D_{1:n_{(1)}}}\big((\hat{\beta}_{(1),k} - \tilde{\beta}_k)^2|n_{(2),k}\big) \\
&\quad + 2\frac{n_{(1),k}n_{(2),k}}{(n_{(1),k} + n_{(2),k})^2}E_{D_{1:n_{(1)}}}\big((\hat{\beta}_{(1),k} - \tilde{\beta}_k)|n_{(2),k}\big)E_{D_{n_{(1)}+1:n}}\big((\hat{\beta}_{(2),k} - \tilde{\beta}_k)|n_{(2),k}\big) \\
&\quad + \frac{n_{(2),k}^2}{(n_{(1),k} + n_{(2),k})^2}E_{D_{n_{(1)}+1:n}}\big((\hat{\beta}_{(2),k} - \tilde{\beta}_k)^2|n_{(2),k}\big)\Big].
\end{aligned}
$$

We have that

$$
E_{D_{n_{(1)}+1:n}}\big((\hat{\beta}_{(2),k} - \tilde{\beta}_k)|n_{(2),k}\big) = 0, \quad E_{D_{n_{(1)}+1:n}}\big((\hat{\beta}_{(2),k} - \tilde{\beta}_k)^2|n_{(2),k}\big) = \frac{\sigma_{Y,k}^2}{n_{(2),k}}
$$

which gives us the desired result.

## 8.10 Proof of Theorem 5.4

By assumption we have that $Y_i$'s are Normally distributed. We first deal with the dependence $E_{D_{1:n_1}}\big((\hat{\beta}_{(1),k} - \tilde{\beta}_{(1),k})^2|n_{(2),k}\big)$. A well known property of the Normal distribution [23] is that the estimate of the mean $\hat{\beta}_{(1),k}$ and the estimate of the variance $\hat{\sigma}_{Y,k}^2$ are independent. This immediately gives that $E_{D_{1:n_1}}\big((\hat{\beta}_{(1),k} - \tilde{\beta}_{(1),k})^2|n_{(2),k}\big) = E_{D_{1:n_1}}\big((\hat{\beta}_{(1),k} - \tilde{\beta}_{(1),k})^2\big) = \frac{\sigma_{Y,k}^2}{n_{(1),k}}$ as there is no dependence between $\hat{\beta}_{(1),k}$ and $n_{(2),k}$. Thus we get that the risk variance for that leaf is just as from Lemma 4.3.

Now we want to bound the probability $\hat{n}_k$ above is far away from $n_k^*$. We will do this by bounding the $a_k$. Another well known property of the normal distribution is $\frac{(n_k-1)S_{Y,k}^2}{\sigma_{Y,k}^2} = (n_k-1)a_k \sim \chi_{(n_k-1)}^2$. By characterization of sub-exponential random variables:

$$P((n_k-1)|a_k-(n-1)| > \sqrt{2(n_k-1)t} + 2t) \le e^{-t}$$

$$P(|a_k-1| > \frac{\sqrt{2t}}{\sqrt{(n_k-1)}} + \frac{2t}{(n_k-1)}) \le e^{-t}$$

$$\frac{\sqrt{2t}}{\sqrt{(n_k-1)}} + \frac{2t}{(n_k-1)} \in (0,1) \implies \frac{2t}{(n_k-1)} \le 1 \implies \frac{2t}{(n_k-1)} < \frac{\sqrt{2t}}{\sqrt{(n_k-1)}}$$

$$\implies P(|a_k-1| > \frac{2\sqrt{2t}}{\sqrt{(n_k-1)}}) \le P(|a_k-1| > \frac{\sqrt{2t}}{\sqrt{(n_k-1)}} + \frac{2t}{(n_k-1)}) \le e^{-t}$$

$$\forall\, \alpha \in (0,1)$$

$$P(|a_k-1| > \alpha) \le e^{-\frac{(n_k-1)\alpha^2}{8}}$$

$$P(\exists k \text{ s.t. } |a_k-1| > \alpha) \le \sum_{k=1}^{K} e^{-\frac{(n_k-1)\alpha^2}{8}}.$$

And now we apply Lemma 4.7 to bound the excess. Now we assume our purely random tree is a Mondrian Tree with the above assumptions, so $n_k = \frac{cn}{K}$. By Markov inequality and Proposition 2 in [20] we have that:

$$P(K_n - 1 > n^{\frac{d+\epsilon}{d+2}}) \le \frac{E[K_n]}{n^{\frac{d+\epsilon}{d+2}}} = \frac{(1+n^{\frac{1}{2+d}})^d}{n^{\frac{d+\epsilon}{d+2}}} = \delta_1$$

$$P(\exists k \text{ s.t. } |a_k-1| > \alpha | K_n \le n^{\frac{d+\epsilon}{d+2}}) \le n^{\frac{d+\epsilon}{d+2}} e^{-\frac{\alpha^2}{8}((cn)^{\frac{2-\epsilon}{d+2}}-1)} = \delta_2$$

By setting $\epsilon = 1$ and using the union bound we get the result.

**Remark.** It is worth noting that in the above proof we have only used the property that $\chi^2$ are subexponetial. A slightly stronger (in terms of $n, \alpha$) inequality is possible using Chernoff Bounds and exploiting the structure of $\chi^2$ random variables.

## 8.11 Dependence in non-normal case

We are interested in the question of when is $E_{D_{1:n_{(1)}}}\big[(\hat{\beta}_{(1),k} - \tilde{\beta}_k)^2 | n_{(2),k} < n_{(2),k}^*\big] < E_{D_{1:n_{(1)}}}\big[(\hat{\beta}_{(1),k} - \tilde{\beta}_k)^2\big]$. Unfortunately $n_{(2),k}$ is a function not only of $\hat{\sigma}_{(1),k}^2$ but of all other $\hat{\sigma}_{(1),l}^2$. Let us start with a more simple and general question of when $E\big[(\hat{\mu}-\mu)^2|\hat{\sigma}^2 < \sigma^2\big] < E\big[(\hat{\mu}-\mu)^2\big]$. We present no formal arguments here but rather share our findings and conjectures which we consider both interesting in their own right as well as excellent candidates for further study. The first observation is that far from this being an unusual property this seems to be a fairly common property. In fact for symmetric distributions the relationship appears to be well behaved. From [23] the sample mean and sample variance are asymptotically MVN (multivariate normal) with cross correlation equal to the skew, so when our distribution is symmetric the sample mean and sample variance are independent in the limit. For the finite sample case the relationship between $\hat{\sigma}_{1,k}^2 - \sigma_{1,k}^2$ and $E\big[(\hat{\mu}-\mu)^2|\hat{\sigma}^2 - \sigma^2\big] - E\big[(\hat{\mu}-\mu)^2\big]$ appear to be monotonic and to go through the origin (so when the sample variance is the true variance, the conditional variance of the sample mean is the unconditioned variance, which is what we would hope is the case). In fact it appears both the magnitude and parity of this relationship depends on the *excess kurtosis* $\kappa - 3$. If $\kappa - 3 < 0$ this relationship is negative and if $\kappa - 3 > 0$ this relationship is positive, with the magnitude increasing as you move further away from zero.

If these observations are true for all symmetric distributions it would be quite fortuitous, since large values of $\kappa$ imply that the estimates of our variances will be more noisy, but those are exactly

the cases where actively fitting to the sample variance of our first stage is beneficial: If our sample variance is larger than the population variance, then the variance of our $\hat{\beta}_{(1),k}$ is larger than expected, so it is beneficial to use more points in the second stage than the optimal passive sampling would have assigned. Meanwhile when a smaller sample variance implies the variance of our $\hat{\beta}_{(1),k}$ is larger than expected, $\kappa$ is small and so our sample variance will itself have small variance. We have not yet been able to prove this relationship, and things become much more complicated in the more realistic case where our distribution is skewed. However these results give us confidence that things are unlikely to go too badly wrong when our labels are not normally distributed.

## 8.12 Experimental data set info

For both simulations our marginal $X$ distribution was uniform over the space $[0,1]^{10}$. Heteroskedastic simulation had constant regression function and Gaussian noise, with space split into high variance region (25) and low variance region (1). Varying complexity had sinusodial regression function $f(x) = C\sin(\frac{2\pi}{d*F} * \sum x_i)$ and Gaussian noise with constant (1) variance. It was split into high variation region ($C = 20, F = 0.05$) and low variation region ($C = 5, F = 0.1$). For both sets $[0.1,1]^{10}$ were the high areas, with everything else a low area.

## 8.13 Practitioners guide

Here we compile information related to actually using this active learning method in practice.

### 8.13.1 Heuristics to deal with difference between theory $n_k^*$ and possible values

There are many reason why you may not actually be able to sample according to your estimates of the optimal $n_k^*$. For a start our $n_k^*$ will almost always be fractional. Additionally there may be less than $n_k^*$ points in a leaf. These issues are fairly minor and become less influential as sample sizes increase. However a more consistent issue that occurs when using the approximating algorithm is when a leaf is oversampled during stage 1, so that $n_{(1),k} > n_k^*$. This means that some other leaf will get fewer than it is optimal number of samples. Although this again can be dealt with asymptotically by making our stage 1 a small fraction of the total number of samples, in practice this is a problem which often occurs when our sample size is not large.

In our code we implemented heuristics to deal with these mismatches. We emphasize that these heuristics are subjective and one could easily use or argue for others. After calculating our $\hat{n}_k$ we immediately floor them all. We then set $\hat{n}_k = \max(\min(\hat{n}_k, \eta_k), n_{1,k})$ (where $\eta_k$ is the total number of points in leaf $k$). It is possible that $\sum \hat{n}_k \neq n$ after these adjustments. If we have too many points, we reduce the largest $\hat{n}_k$ until we achieve the correct total. If we have too few points we increase the $\hat{n}_k$ by 1 each, starting with the smallest, and starting over once we have increased them all by 1. This asymmetry is because increasing small values can have a large reduction on the variance of the estimate, but decreasing large ones leads to a small increase in variance.

### 8.13.2 Lifetime parameter sequence

We have found that the best general form for the lifetime parameter sequence is $\lambda_n = \frac{1}{\gamma}(n^{\frac{2}{2+d}} - 1)$. The $\gamma$ can be fairly freely chosen with $\gamma = 1$ a reasonable default (and is what is used in all simulations and experiments in this paper), but the $-1$ is very important; it ensures that we do not start with a lifetime $= 1$ for $n = 1$, $\forall d$ as when $d$ is large this can result in a very large number of leaves early on.

### 8.13.3 Sampling method during stage 1

During stage one our theory assumed that $n_1 = cn$ and then each leaf received the same fraction of points, as this gives important asymptotic properties. In practice if $c$ is too large this can result in putting too many samples in certain small leaves during stage 1, so that $n_{1,k} > n_k^*$, meaning that we have oversampled this leaf and will have to reduce other sampling elsewhere. One way of avoiding this is by making $c$ small, but this risks getting bad leaf estimates and suboptimal stage 2 sampling unless $n$ is large, where the $n$ required increases as $d$ increases. Another is to sample passively. We have found that generally if $c = 0.5$ then sampling passively tends to produce pretty good results

unless your function has massive amounts of variation. Another option is to use a hybrid sampling scheme in stage 1, where each leaf is given a small number of samples, and then the rest of the samples are distributed randomly, but empirically this seems to be worse than random sampling for small values of $n$.

### 8.13.4 Final regression model

As shown in our experiments, although most the theory assumes that you are using the same tree for your active learning as you are for your final predictions, you also get good results doing active learning with Mondrian Forests, and then taking that data and fitting your final model with a more data adaptive model, although not always.

### 8.13.5 Forests

Just as with Breiman decision trees you can ensemble purely random trees into forests. These forests show improved performance at the cost of increased computational cost since they average out the random process used to build the trees. We also have an intuitive (though theory free) extension of our active learning method to utilize the power of multiple Mondrian Trees. The idea is each tree determines the optimal number of samples per leaf in the usual way, and then gives data points weights such that the expected number of points sampled from each leaf is the optimal number. These probabilities are then averaged out over all the trees in the forest and the new points are sampled using these averaged probabilities. The formal algorithm is given below:

---

**Algorithm 3:** Forest version of oracle approximation algorithm

**Input:** Leaves of our $T$ trees $\mathcal{I}_1...\mathcal{I}_T$, pool of data points $\{X_i\}_{i=1}^m$, and label budgets
    $n_{(1)}, n_{(2)}, n = n_{(1)} + n_{(2)}$.
**Output:** The set of labelled points.
*Stage 1:* ;
Sample $n_{(1)}$ data points (possibly according to the structure of the trees $\mathcal{I}_t$) using a version of
  algorithm 1. ;
**foreach** $t$ **do**
  | Use those samples $(X_i, Y_i)$ to estimate $\hat{\sigma}_{Y,k,t}^2$ for each leaf. ;
**end**
*Stage 2:* ;
**foreach** $t$ **do**
  | **foreach** $I_{k,t} \in \mathcal{I}_t$ **do**
  | | Calculate $\hat{n}_{k,t} = n \frac{\sqrt{p_{k,t}\hat{\sigma}_{Y,k,t}^2}}{\sum_{k'} \sqrt{p_{k',t}\hat{\sigma}_{Y,k',t}^2}}$ the number of points in the leaf to sample. ;
  | | Count $m_{k,t}$ the number of unlabelled points in leaf $I_{k,t}$ ;
  | | **foreach** *Unlabelled* $X_i \in I_{k,t}$ **do**
  | | | Assign weight $W_{i,t} = \frac{\hat{n}_{k,t} - n_{(1),k,t}}{n_2 * m_{k,t}}$. ;
  | | **end**
  | **end**
**end**
**foreach** *Unlabelled* $X_i$ **do**
  | Final weight $W_i = \frac{1}{T} \sum W_{i,t}$. ;
**end**
Sample $n_{(2)}$ points with weights $W_i$.

---

Below we show the results of using Mondrian Forests for our active learning, and both Mondrian Forests and Random Forests as our final regression model. Here we see some benefit using Mondrian Forest for active learning and then Random Forests for our final regressor (although in fact the naive uncertainty sampling method outperforms ours). Although the benefit on the real data appears to be a small constant factor, the actively learned models provide similar accuracy with 10s of fewer data points, which can be significant.

Figure 2: Mondrian Forest active learning simulations

#### 8.13.6 Using more than 2 stages

It is of course possible to do more than 2 stages, updating your estimates of the leaf variances during each stage to guide sampling during the next stage. We found that in practice the benefits of doing this are generally fairly small. Of course the first stage should still be sufficiently large that you get decent initial estimates for the leaf variances. Much of the theory could be extended to increasing number of stages as long that the number is not increasing with $n$ without much work. Increasing the number of stages as $n$ increases may require additional care and effort.

Figure 3: Mondrian Forest active learning experiments

### 8.13.7 Additional experimental results

Figure 4: Additional active learning experiments on UCI data with Mondrian Trees