[Reviews · NeurIPS 2018]

Reviewer 1



•Post review comments: Assuming the authors will indeed address the unclear parts that I indicated, and in particular the proof of cor. 4.6 I am updating my score. Yet, I still think that their results (including those appearing in their response) are not showing significant gains as seen in other Active Learning settings. The paper addresses the problem of active learning for regression with random trees. The authors introduce a simple ‘oracle’ algorithm and a risk variance is provided for this criterion. Authors provide a risk minimizing query criteria and claim it is better than the random selection criterion An estimation algorithm is presented that provides the necessary statistics at its first stage to allow an approximation to the ‘oracle’ algorithm. Numerical results are presented in which improved performance over random and uncertainty sample is provided for simulated data, and no improvement for real UCI data. Strengths: • The paper provides both an algorithm and theoretical analysis with empirical results. Theoretical analysis for active learning algorithms is a challenging task, lacking in many of the state-of-the-art heuristic algorithms for active learning • The organization of the paper’s sections is good Weaknesses: 1. There are many unclear parts in the paper, lacking formalism in some places which leaves the reader confused. I will give some examples for that in the order in which they appear a. Introduction, 2nd paragraph – the example requires the authors to make some analogy between the chemical quantities of the systems and concepts in active learning. It is completely unclear to me how this example makes a case for active learning. b. The authors make first reference to a leaf in the setting and background in page 2 line 54. However, no definition of a tree has been made yet, or even a simple reference to such. c. Authors refer to n and d in line 93 and in algorithm 1 however it was never made clear what they are. ‘n’, for example: Is ‘n’ the total size of data? Is it the test set size? Pool set size? d. Preposition 4.1 is not clear are we looking at the post selected leaf-set X after q_k proportion was selected? e. Most important: Key Corollary 4.6 proof is unclear, short, informal and confusing: what minimum and average are your referring to? This is a key result of the paper and it should be very clearly explained and proved! 2. Author should explain and motivate their decision to let $a_k=\frac{p_k}{\sgima^2_{Y,k}} in line 156. Why should the multiplicative factor have such a form? 3. Experimental results: Results are not convincing in demonstrating that the method is advantageous for real data. Authors should provide such examples or improve their algorithm. 4. In addition: a. Authors should provide basic data sizes for the datasets they experiment with in order for the reader to evaluate the efficiency of their method by, for example, comparing the size of the training set with the size of the full data set. b. Please provide reference for Breiman trees 5. Minor: many typos exist, authors should proof read their paper.

Reviewer 2



This paper deals with the problem of active learning using random trees for the case of regression. The approach proposes a technique for selecting candidate points for updating the trees, which can be a standard random tree or a Mondrian tree. The approach is tested on the simulation and real data. In general, I enjoyed reading this paper. It is nicely written, with very clear statements and thorough analysis. The structure of the paper is clear making it quite easy to read and follow. The contribution of this paper is mainly on the theoretical side, while providing very nice insights. On the practical side, it is not clear how useful this proposed approach would be. First, learning a regression function using trees might not be the best thing to do. Due to the tree properties, generalization on unknown data points might be difficult. Especially, when doing prediction, jumping between different leaves might result into non-smooth behavior, limiting the usage of this kind of approach. Secondly, it might scale badly with dimensions. Furthermore, it is not clear to me how robust this kind of approach towards noise is. These might explain the sub-optimal results when going for real-world data, as shown by the evaluation, where the proposed active learning does not performed better than the random selection scheme.

Reviewer 3



I have read reviewers feedback and will keep my original comments. This paper studies active learning for non-parametric regression for random trees. The proposed algorithm can suggest the number of new samples to acquire in each branch of the tree. The considered problem seems to be timely and important, since indeed, as the author commented, actively learning has been heavily focused on binary classification. The paper is generally well-written. The paper derived the prediction error for tree regression (which entails the classic bais-variance tradeoff). Then the *number of samples* needed to take for each branch to minimize the average prediction error is derived in closed-form by solving a simple optimization problem. An active learning algorithm is proposed based on this. It seems to me that the main result is this optimal *number of samples* for each leaf node, subject to a total sample constraint, such that the overall prediction error is minimized. Theorem 5.1 seems to the main theoretical result, which derives the consistency of the sampling. This gives a specific performance guarantee. Simulations are provided to verify the results. There is no real-data example - it could significantly enhance the paper. As a theory paper, it contains some interesting formulation and results (which are moderately significantly and challenging). Thus I would like to suggest a marginal acceptance.